

# Artificial light at night alters life history in a nocturnal orb-web spider

Nikolas J. Willmott, Jessica Henneken, Caitlin J. Selleck and
Therésa M. Jones

School of BioSciences, University of Melbourne, Melbourne, VIC, Australia

## ABSTRACT

The prevalence of artificial light at night (ALAN) is increasing rapidly around the world. The potential physiological costs of this night lighting are often evident in life history shifts. We investigated the effects of chronic night-time exposure to ecologically relevant levels of LED lighting on the life history traits of the nocturnal Australian garden orb-web spider (*Eriophora biapicata*). We reared spiders under a 12-h day and either a 12-h natural darkness (~0 lux) or a 12-h dim light (~20 lux) night and assessed juvenile development, growth and mortality, and adult reproductive success and survival. We found that exposure to ALAN accelerated juvenile development, resulting in spiders progressing through fewer moults, and maturing earlier and at a smaller size. There was a significant increase in daily juvenile mortality for spiders reared under 20 lux, but the earlier maturation resulted in a comparable number of 0 lux and 20 lux spiders reaching maturity. Exposure to ALAN also considerably reduced the number of eggs produced by females, and this was largely associated with ALAN-induced reductions in body size. Despite previous observations of increased fitness for some orb-web spiders in urban areas and near night lighting, it appears that exposure to artificial night lighting may lead to considerable developmental costs. Future research will need to consider the detrimental effects of ALAN combined with foraging benefits when studying nocturnal insectivores that forage around artificial lights.

## INTRODUCTION

The spread of artificial light at night (ALAN) is increasing rapidly around the globe and its presence has been linked to shifts in physiological and behavioural traits in animals (*Gaston et al., 2013*; *Longcore & Rich, 2004*). Exposure to ALAN is directly linked to changes in key life history traits, including variation in patterns of juvenile growth (*Brüning, Hölker & Wolter, 2011*), reductions in immune function (*Bedrosian et al., 2011*; *Durrant et al., 2015*), survival (*Shah et al., 2011*) and fecundity (*McLay, Green & Jones, 2017*), as well as shifts in reproductive behaviours (*Firebaugh & Haynes, 2016*; *McLay, Green & Jones, 2017*; *Van Geffen et al., 2014*, *2015*). Indirectly, where exposure to night lighting stimulates earlier maturation and smaller adult size, there may be reproductive costs due to poorer male performance (*Elgar & Jones, 2008*; *Van Geffen et al., 2014*) and reduced female fecundity (*Honěk, 1993*; *Van Geffen et al., 2014*). Adults that emerge

Corresponding author
Nikolas J. Willmott,
nwillmott@student.unimelb.edu.au

earlier may benefit if this extends their potential breeding period and thus increases offspring production (*Lowe, Wilder & Hochuli, 2014*; *Schneider & Elgar, 2002*). However, each sex may respond differently to the presence of ALAN, and maturation time for males and females could desynchronise, potentially resulting in catastrophic reductions in reproductive success (*Dominoni, Quetting & Partecke, 2013*; *Van Geffen et al., 2014*).

A potential underlying mechanism for these life history shifts is that the presence of ALAN may simulate a longer day and/or mask daily and seasonal patterns of light. Changes in natural lighting drive a multitude of daily and seasonal biological rhythms, including reproduction (*Nelson, Denlinger & Somers, 2010*; *Tauber, Tauber & Masaki, 1986*), and thus ALAN may have serious consequences for juvenile development and reproductive maturation (*McLay, Green & Jones, 2017*; *Shah et al., 2011*; *Van Geffen et al., 2014*). For species that exhibit seasonal diapause or overwintering (*Schaefer, 1987*; *Shah et al., 2011*; *Van Geffen et al., 2014*), disruption by ALAN (simulating long-day photoperiods that naturally suppress diapause; *Nylin & Gotthard, 1998*) can result in earlier maturation (*Van Geffen et al., 2014*) and may increase mortality (*Shah et al., 2011*). However, there is variation in these patterns across taxa, as development time may also be shortened in vertebrate species that do not exhibit diapause (*Brüning, Hölker & Wolter, 2011*; *Dominoni, Quetting & Partecke, 2013*). This suggests the effects of ALAN on life history depend on species-specific physiological and life history traits.

The detrimental effects of ALAN are potentially exacerbated if animals are attracted to artificial lights. Many species exhibit positive phototaxis (e.g. attraction towards natural moonlight or light reflecting off water bodies), which is an adaptive trait facilitating behaviours such as navigation (*Minnaar et al., 2015*; *Van Langevelde et al., 2011*) or foraging site choice (*Heiling, 1999*; *Rydell, 1992*). For these species, ALAN may act as an ecological trap (*sensu*, *Hale & Swearer, 2016*) resulting in individuals settling in poor quality habitats (*Eisenbeis & Hänel, 2009*; *Gaston et al., 2012*; *Longcore & Rich, 2004*). Conversely, nocturnal insectivores may gain direct benefits if they forage around artificial lights because increased prey densities around these lights facilitate increased foraging success (*Adams, 2000*; *Heiling & Herberstein, 1999*; *Lacoeuilhe et al., 2014*). However, while research demonstrates that insectivores (in particular) may gain foraging benefits from the presence of ALAN, the potential physiological costs of ALAN are largely unstudied.

Spiders are an ideal taxon for investigating the relative ecological costs and benefits of the presence of ALAN. Simulated long-day photoperiods may induce shifts in a spider's maturation period, reducing the number of juvenile instars (*Miyashita, 1987*; *Nylin & Gotthard, 1998*; *Schaefer, 1987*), a pattern that likely varies with species and life history stage (*Schaefer, 1987*). Additionally, some species are urban exploiters, meaning they perform well in urban habitats (*Bolger et al., 2008*; *Lowe, Wilder & Hochuli, 2014*; *Shochat et al., 2004*). However, it is not always clear which urbanisation factors (e.g., light, noise, temperature and habitat fragmentation) or which species traits drive this urban exploitation (*Trubl et al., 2012*). Broadly, urbanisation drives shifts in temperature, prey availability, and other correlated factors that potentially alter spider development, including time to maturation and total growth (*Bonaric, 1987*;

*Lowe, Wilder & Hochuli, 2014*; *Mayntz, Toft & Vollrath, 2003*; *Vollrath, 1987*). The specific impact of artificial light is often omitted when investigating the effects of urbanisation on spiders (*Dahirel et al., 2017*; *Lowe, Wilder & Hochuli, 2014*). This is a surprising oversight, as many orb-web spiders preferentially construct their webs in open spaces and so are likely to be attracted to artificial lighting during foraging site choice (*Craig & Bernard, 1990*; *Heiling, 1999*), and importantly, artificial lights aggregate their nocturnal insect prey (*Heiling & Herberstein, 1999*; *Kreiter & Wise, 2001*; *Longcore & Rich, 2004*). Additionally, orb-web spiders are particularly immobile predators once they have selected a foraging site, so compared to more mobile insectivores such as bats and geckos, they are likely to be exposed to ALAN to a much greater extent.

We investigated the effects of lifetime exposure to ALAN on key life history traits (juvenile development rate, total growth and survival, as well as adult reproductive output and survival) in the Australian garden orb-web spider (*Eriophora biapicata*). These large nocturnal spiders (body length up to 22 mm in females and 18 mm in males) are prevalent in urban and suburban habitats and often build their webs near or on artificial lights. Additionally, they forage primarily on species of Lepidoptera, Coleoptera, and Diptera (*Herberstein & Elgar, 1994*), many of which are highly attracted to lights (*Eisenbeis & Hänel, 2009*). In good quality foraging sites such as around street lights, which attract abundant nocturnal invertebrate prey, *E. biapicata* juveniles demonstrate high foraging site persistence (N. J. Willmott et al., 2016–2017, unpublished data). Hence, they are an ideal species for investigating the life history consequences of ALAN exposure, as they are likely to be chronically exposed to the effects of ALAN in urban areas.

# MATERIALS AND METHODS

## Collection and housing

Experimental spiders were obtained from eggsacs laid in the laboratory by 18 wild-caught *E. biapicata* females, collected from sites ranging in light intensity from <0.1 to 40 lux (Skye Instruments Lux Meter, Llandrindod Wells, Wales, UK) in an urban park in Melbourne, Victoria (37.7911 S, 144.9515 E) in February 2016. Spiderlings from these 18 families were reared from hatching at 22 °C under a 12-h day (2000 lux; 12 V cool white LED strip lighting; Fig. S1) and a 12-h night that was either darkness (*0 lux* treatment; *n* = 215; 0–0.06 lux) or dim light at night (*20 lux* treatment; *n* = 235; 20–24.6 lux; 12 V cool white LED strip lighting). We selected *20 lux* as it sits in the middle of the range of light intensities experienced by spiders in our collection site, and is comparable to other laboratory studies mimicking the effects of street lighting (*Durrant et al., 2018*; *McLay, Green & Jones, 2017*). Offspring from each family contributed equally to each of the two light treatments. At 14.95 ± 1.47 (mean ± SE) days after emergence from the egg sac, juvenile spiders were transferred to inverted, transparent plastic cups and maintained until death under standard laboratory conditions (*Henneken et al., 2015*). To maintain humidity, cups were lightly misted with water every two days. Young juveniles (hatching until seventh instar) were provided with three to five *Drosophila melanogaster* per week; older juveniles (seventh instar to penultimate instar) were fed three to five house flies (*Musca domestica*) per week; and, adults were fed one juvenile cricket (*Acheta domesticus*)

equal to their body size twice a week. All spiders were provided with equal food appropriate to their size and life stage regardless of treatment group. The amount of food provided to juveniles was based on observations of wild *Eriophora*. We did not provide ad libitum food during development, which would have been unrealistic and may have masked the effects of ALAN. Spiders were handfed where possible to minimise the effects of lights on prey capture rates. Prey were rarely left completely uneaten regardless of treatment group but adult spiders often discarded food suggesting that they were likely provided ad libitum food.

## Maturation time, survival, and body size

Spiders were checked every two to three days and any moults or deaths were noted, to assess the effects of the presence of ALAN on development rates and survivorship. The date of death after maturation was taken as a measure of adult lifespan. The age (in days) at which a spider completed its final moult was defined as its age at maturation. Once spiders reached maturity, we measured their body mass (mg) and the length of the tibia (mm) on the front left leg, and used these to determine total life-time growth.

## Reproductive success

We assessed whether exposure to ALAN during development affected reproductive success by providing spiders with mating opportunities within their own light treatment groups and measuring the number of offspring produced. Mating pairs were age-matched to ensure senescence (days since maturation) differed as little as possible between treatment groups (mean ± SEM senescence at mating: *0 lux* males = 20.32 ± 2.36; females = 8.14 ± 0.81; *20 lux* males = 18.96 ± 2.56; females = 10.40 ± 2.17). Prior to each mating trial, we allowed the female to build a web in a rectangular Perspex frame (58 × 58 × 15 cm) after 'sunset' in the laboratory. Following web construction, we placed the male in the bottom corner of the frame furthest from the hub of the female's web to allow mating. Following mating, both males and females were returned to their normal housing conditions. If a female failed to mate with a male, she was paired with consecutive different males until she mated. For each mated female, we recorded the time between mating and production of the first eggsac, the total number of eggsacs laid, the number of spiderlings that emerged, and the average mass of individual spiderlings in the first eggsac produced by each female; spiderling mass in subsequent eggsacs were not measured due to time constraints. To measure reproductive costs of ALAN due to physiological disruption, independent of reproductive costs due to shifts in body size, we calculated the number of eggs produced per milligram of female body mass for each mated female.

## Statistical analysis

Statistical analyses were carried out using R version 3.4.2 (*R Core Team, 2017*). We used linear mixed models to determine the effects of light treatment (fixed factor), sex (fixed factor) and family (random factor) on age at maturation, length of the intermoult period, and the number of recorded moults. We used a generalised linear mixed model

(with an assumed logit link) to compare the proportion of juveniles in each treatment group surviving to maturity, with family as a random factor. We used linear mixed models to determine the effects of light treatment (fixed factor), sex (fixed factor) and family (random factor) on body mass, and tibia length in adults. In the case of a significant treatment × sex interaction, we used post hoc comparisons with a Bonferroni correction. Due to the strong correlation between treatment group and age at maturation, relationships between age at maturation and body mass, tibia length, or number of juvenile moults were analysed separately for each treatment group. The marginal ($R^2_m$) and conditional ($R^2_c$) $R^2$ values are presented for these analyses. Survival curves (for entire lifespan and separately for adult longevity only) for each treatment group were built using a Kaplan–Meier survival analysis (lifespan or adult longevity as response, light treatment as fixed factor, family as random factor; sex was a fixed factor for the analysis of adult longevity). Curves were compared using a log-rank test (survival package in R; *Therneau, 2015*). Linear mixed models were used to test the effects of light treatment (fixed factor) and family (random factor) on the number of eggsacs produced, the number of spiderlings per eggsac, the time to the first eggsac, and the mass of spiderlings in the first eggsac.

## RESULTS

### Maturation time and survival

The probability that a juvenile reached maturity was comparable for the two light treatments (72 of 215 *0 lux* spiderlings; 64 of 235 *20 lux* spiderlings; GLMM: $\chi^2$ = 2.54, d$f$ = 1, $P$ = 0.11). However, *20 lux* spiders matured significantly earlier (Table 1A) and required fewer moults to reach maturity (Table 1B) compared with *0 lux* spiders, for both males and females. The intermoult interval (days) was comparable for *20 lux* and *0 lux* spiders (Table 1C). There was a positive relationship between time to maturation and the recorded number of moults ($F_{(1, 131)}$ = 109.1, $P$ < 0.0001; Fig. 1). Overall, the total number of days survived (from emergence from the egg to death) by *20 lux* spiders was fewer than *0 lux* spiders (log-rank test: $\chi^2$ = 10.90, d$f$ = 1, $P$ < 0.001; Fig. 2) but adult survival was comparable for the two light treatments ($\chi^2$ = 0.001, d$f$ = 1, $P$ = 0.97). Regardless of lighting treatment, males matured earlier than females (Table 1A) and required fewer moults to reach maturity (Table 1B). The intermoult interval (days) was shorter for females compared to males (Table 1C) and adult females lived longer than males ($\chi^2$ = 41.85, d$f$ = 1, $P$ < 0.0001; Fig. 3).

### Body size

There was a significant interaction between light treatment and sex in their effect on body mass at maturation (Table 1D). Post hoc analyses revealed that female body mass was significantly greater than male body mass in the *0 lux* group (estimate ± SE: 169.88 ± 19.05; $t_{130.6}$ = 8.92, $P$ < 0.0001), but not in the *20 lux* treatment group (estimate ± SE: 36.32 ± 19.98; $t_{124.3}$ = 1.82, $P$ = 0.29). Additionally, *20 lux* spiders were significantly smaller than *0 lux* spiders, with a bigger effect in females (males: estimate ± SE: 158.40 ± 18.14; $t_{131.4}$ = 8.73, $P$ < 0.0001; females: estimate ± SE: 291.97 ± 20.70; $t_{124.9}$ = 14.11, $P$ < 0.0001). There was also a significant treatment–sex interaction for tibia length

**Table 1 Effects of light treatment on development time and total growth.**

| Models | 0 lux at night | | 20 lux at night | | Statistic | P-value |
|---|---|---|---|---|---|---|
| | Males | Females | Males | Females | | |
| (a) Age at maturation (days) | | | | | | |
| Light treatment | 300.28 ± 6.22 | 334.91 ± 7.12 | 262.03 ± 6.11 | 272.35 ± 5.38 | $F_{(1, 131)} = 85.78$ | **<0.0001** |
| Sex | | | | | $F_{(1, 131)} = 16.97$ | **<0.0001** |
| (b) Number of juvenile moults | | | | | | |
| Light treatment | 11.61 ± 0.30 | 12.90 ± 0.36 | 10.33 ± 0.29 | 11.17 ± 0.34 | $F_{(1, 131)} = 25.28$ | **<0.0001** |
| Sex | | | | | $F_{(1, 131)} = 10.99$ | **0.001** |
| (c) Length of intermoult period (days) | | | | | | |
| Light treatment | 31.30 ± 1.17 | 30.42 ± 0.80 | 33.05 ± 1.62 | 28.01 ± 1.01 | $F_{(1, 131)} = 0.02$ | 0.89 |
| Sex | | | | | $F_{(1, 131)} = 4.91$ | **0.03** |
| (d) Body mass (mg) | | | | | | |
| Light treatment | 290.1 ± 14.44[a] | 456.07 ± 19.78[b] | 126.47 ± 8.70[c] | 162.56 ± 7.99[d] | $F_{(1, 131)} = 247.6$ | **<0.0001** |
| Sex | | | | | $F_{(1, 131)} = 57.15$ | **<0.0001** |
| Light treatment × sex | | | | | $F_{(1, 131)} = 23.32$ | **<0.0001** |
| (e) Tibia length (mm) | | | | | | |
| Light treatment | 6.98 ± 0.14[a] | 6.81 ± 0.11[a] | 5.43 ± 0.11[b] | 4.78 ± 0.10[c] | $F_{(1, 131)} = 163.16$ | **<0.0001** |
| Sex | | | | | $F_{(1, 131)} = 0.52$ | 0.47 |
| Light treatment × sex | | | | | $F_{(1, 131)} = 10.71$ | **0.001** |

**Notes:**
Measures (mean ± SE) of development time and total growth at maturity for males and females in the two lighting treatment groups—dark at night (0 lux) and light at night (20 lux), and the full models for these effects. Non-significant ($P > 0.1$) interactions were dropped from models. Superscript letters above means and standard errors denote significant differences ($P < 0.05$) in treatment-specific or sex-specific comparisons.
Significant P-values ($P < 0.05$) are shown in bold.

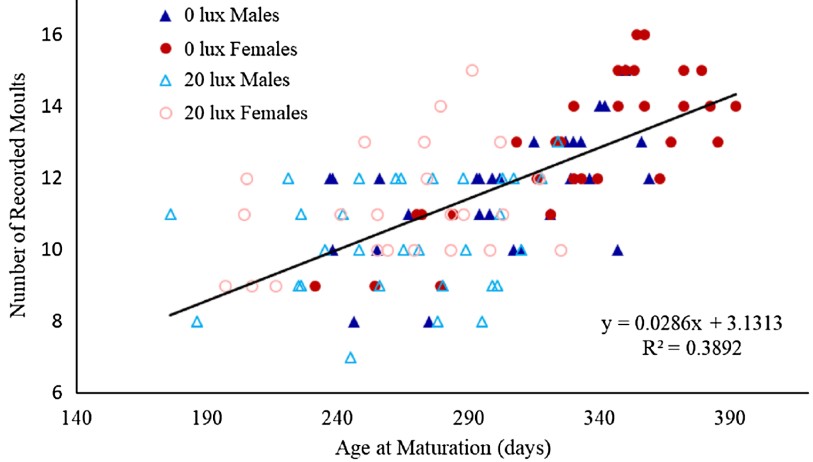

**Figure 1 Relationship between number of moults and age at which spiders reached final moult.** The relationship between the number of recorded moults and age (in days) at which male and female spiders reared in the dark at night (0 lux) and light at night (20 lux) treatments completed their final moult. The black line represents the fitted regression line ($y = 0.03x + 3.13$; $R^2_m = 0.48$ $R^2_c = 0.61$) for the overall relationship between number of moults and age at maturation.

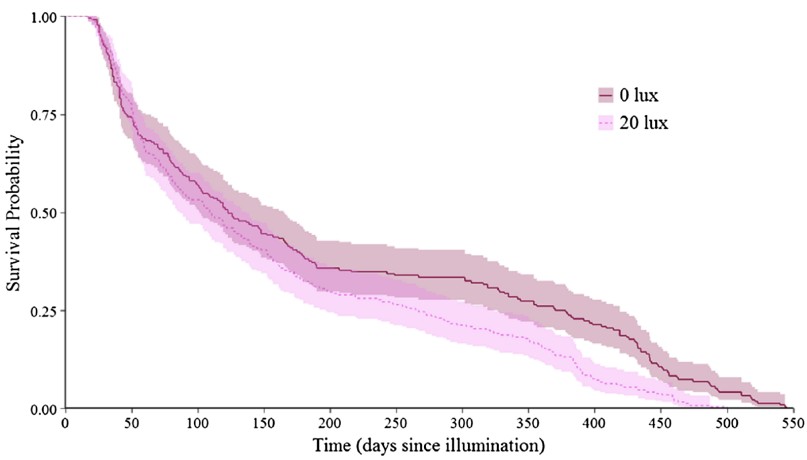

**Figure 2 Effect of light treatment on survival over the entire lifespan.** Survivorship curve for dark at night (0 lux) and light at night (20 lux) treatment group spiders over their entire lifespan. Day 0 represents 14.95 ± 1.47 (mean ± SE) days since hatching, at which point spiderlings were placed individually into cups and subjected to their lighting conditions. Prior to this point, all spiderlings experienced dark at night conditions. Shaded areas represent 95% confidence intervals from a Kaplan–Meier survival analysis.

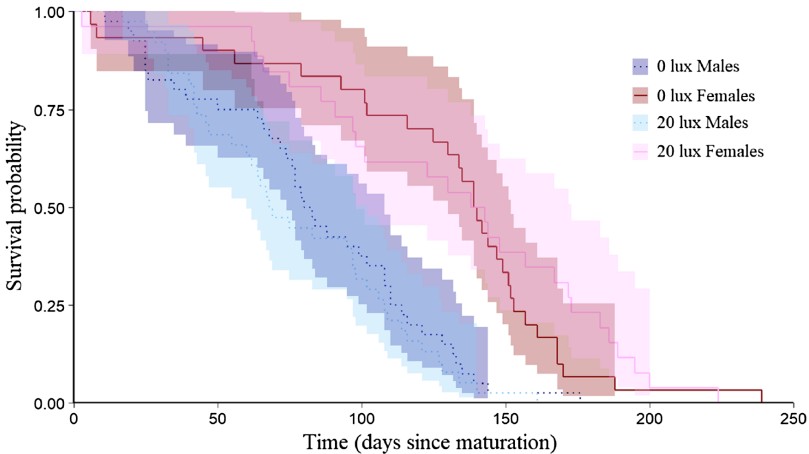

**Figure 3 Effect of light treatment on adult survival.** Adult longevity (survival probabilities as adults) curves for dark at night (0 lux) and light at night (20 lux) treatment males and females. Day 0 represents the day at which each spider matured. Shaded areas represent 95% confidence intervals from a Kaplan–Meier survival analysis.

(Table 1E). Post hoc analyses demonstrated that male tibia length was greater than female tibia length for *20 lux* spiders (estimate ± SE: 0.66 ± 0.17; $t_{124.7}$ = 3.91, $P < 0.001$) but the sexes were comparable in *0 lux* spiders (estimate ± SE: 0.18 ± 10.16; $t_{130.2}$ = 1.10, $P = 1.00$). Tibia length was significantly greater in *0 lux* compared to *20 lux* spiders in both males (estimate ± SE: 1.51 ± 0.15; $t_{131.2}$ = 9.79, $P < 0.0001$) and females (estimate ± SE: 2.00 ± 0.18; $t_{125.4}$ = 11.37, $P < 0.0001$). There was a positive relationship between time to maturation and adult body mass ($F_{(1, 131)}$ = 262.35, $P < 0.0001$; Fig. 4) and adult tibial length ($F_{(1, 131)}$ = 40.55, $P < 0.0001$; Fig. 5).

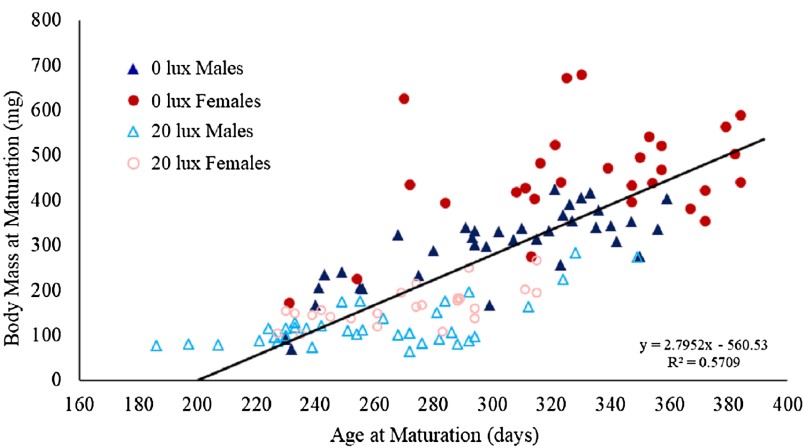

**Figure 4 Effect of light treatment and sex on the relationship between body mass and age at maturity.** The relationship between body mass (mg) at maturity (after final moult and before feeding again) and age (in days) at which male and female spiders reared in the dark at night (0 lux) and light at night (20 lux) treatments completed their final moults. The black line represents the fitted regression line ($y = 2.76x - 560.33$; $R_m^2 = 0.61$, $R_c^2 = 0.75$) for the overall relationship between body mass and age at maturation.

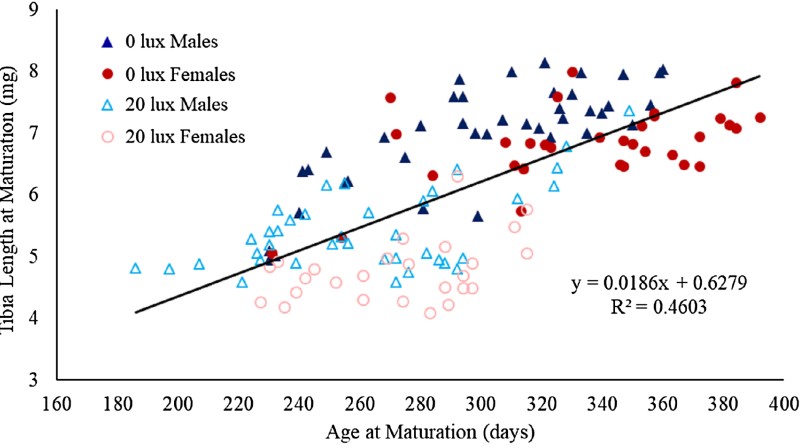

**Figure 5 Effect of light treatment and sex on the relationship between tibial length and age at maturity.** The relationship between tibial length (mm) at maturity and age (in days) at which male and female spiders reared in the dark at night (0 lux) and light at night (20 lux) treatments completed their final moults. The black line represents the fitted regression line ($y = 0.02x + 0.63$; $R_m^2 = 0.49$, $R_c^2 = 0.65$) for the overall relationship between tibial length and age at maturation.

## Reproductive success

There was no effect of light treatment on the likelihood of a given pair mating ($F_{(1, 79)} = 1.34$, $P = 0.25$). The number of eggsacs laid per female was comparable across light treatments (Table 2A), but *20 lux* females produced fewer spiderlings overall (Table 2B). On average, *20 lux* females produced 63.96% fewer spiderlings per eggsac (main effect: $\chi^2 = 218.35$, df = 1, $P < 0.0001$; Table 2C). The number of spiderlings per eggsac declined with eggsac number (main effect: $\chi^2 = 30.07$, df = 1, $P < 0.0001$), doing so faster in *0 lux* females (interaction: $\chi^2 = 8.42$, df = 1, $P < 0.004$; Fig. 6). The egg to mass ratio (the ratio of the

**Table 2 Effects of light treatment on fecundity and offspring size.**

| Models | 0 lux at night | 20 lux at night | Statistic | P-value |
|---|---|---|---|---|
| (a) Eggsacs per female (count) | | | | |
| Light treatment | 5.18 ± 0.47 | 5.40 ± 0.46 | $\chi^2 = 0.11$, $df = 1$ | 0.74 |
| (b) Spiderlings per female (count) | | | | |
| Light treatment | 4,367.92 ± 361.61 | 1,471.27 ± 234.37 | $\chi^2 = 43.06$, $df = 1$ | **<0.0001** |
| (c) Spiderlings per eggsac (count) | | | | |
| Light treatment | 895.07 ± 35.01 | 311.23 ± 22.33 | $\chi^2 = 183.34$, $df = 1$ | **<0.0001** |
| (d) Individual spiderling mass (mg) | | | | |
| Light treatment | 2.36 ± 0.05 | 2.33 ± 0.05 | $F_{(1, 45)} = 0.16$ | 0.70 |

**Note:**
Measures (mean ± SE) of fecundity and offspring size for spiders in the dark at night (0 lux) and light at night (20 lux) treatments.
Significant P-values ($P < 0.05$) are shown in bold.

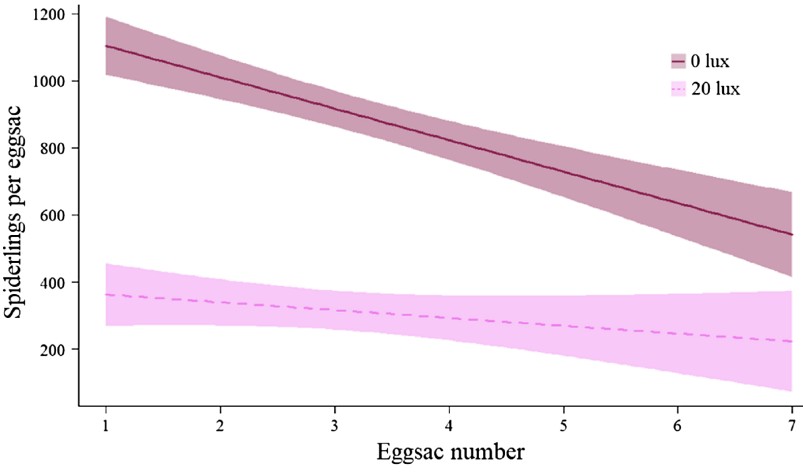

**Figure 6 Effect of light treatment on the number of offspring produced by females.** The relationship between the number of spiderlings per eggsac and the eggsac number for females in the dark at night (0 lux) and light at night (20 lux) treatments. Shaded areas represent 95% confidence bands.

number of eggs per eggsac to female body mass at maturity) was not significantly different between treatment groups (0 lux = 2.11 ± 0.12; 20 lux = 1.84 ± 0.16 eggs per mg of female; $t_{45} = 1.33$, $P = 0.19$). The time between mating and laying the first eggsac was comparable between treatment groups (mean ± SEM days: 0 lux = 28.54 ± 2.54; 20 lux = 22.92 ± 2.15; $F_{(1, 49)} = 2.83$, $P = 0.1$). There was no significant effect of light treatment on the mass of individual spiderlings in the first eggsac (Table 2D).

# DISCUSSION

Our study demonstrated dramatic shifts in key life history traits of *E. biapicata* resulting from chronic exposure to ALAN. Exposure to ALAN reduced the number of juvenile instars, which resulted in earlier maturation at a smaller body size and ultimately led to a significant reduction in reproductive output. Mortality rates were higher in spiders exposed to ALAN, although earlier maturation meant there was no significant difference

between treatment groups in the proportion that reached maturity. Here, we discuss the consequences of these life history shifts in the context of increased foraging success for spiders that build their webs near artificial lights, as we predict this may counter the physiological costs observed.

## Effects of ALAN on spider life history traits

Seasonal shifts in temperature and daily photoperiod regulate patterns of growth and development in the majority of animals, through underlying effects on circadian and circannual rhythms (*Adkisson, 1966*; *Gaston et al., 2013*; *Fonken & Nelson, 2014*; *Navara & Nelson, 2007*). Spiders exposed to ALAN, which may emulate the longer photoperiod normally associated with spring and summer, progressed through fewer juvenile moults and matured earlier. This is consistent with similar observations of extended photoperiods reducing instar number and stimulating earlier maturation in the house spider *Achaearanea tepidariorum* Koch 1841 (*Miyashita, 1987*). Similarly, ALAN exposure shortened seasonal diapause and caused earlier maturation in other arthropod species (*Shah et al., 2011*; *Van Geffen et al., 2014*), and overwintering as early instar juveniles has been documented for several spider species (*Schaefer, 1987*). One possible explanation is that spiders exposed to ALAN may skip several winter moults as the nocturnal photic cues they experience mimic spring and summer photoperiods. In our study, we did not observe an overwintering period during which spiders did not moult for either treatment group. Variation in the number of moults required to reach maturity in spiders has been attributed to the photoperiod experienced (*Miyashita, 1987*). Spiders such as *Eriophora* may exhibit this plasticity if the ultimate moult is related to when their sexual organs are sufficiently developed, rather than moult number. Hence, perception of an extended photoperiod (indicating the longer days typical of the reproductive season) would stimulate faster sexual development and earlier maturation, potentially driven by a fundamental shift in the biological clocks of the spiders. It is possible that moulting occurs during natural overwintering, but these moults are likely associated with only small increases in body mass. This shift in the rate of development may be related to a documented physiological effect of ALAN: its suppression of the nocturnal synthesis of melatonin (*Jones et al., 2015*). Melatonin is a highly evolutionarily conserved molecule (*Vivien-Roels & Pévet, 1993*) that is linked to circadian and circannual rhythms and appears to have a regulatory function for moulting in arthropods (*Girish, Swetha & Reddy, 2015*). However, the functions of melatonin in spiders remain largely untested.

Earlier maturation has potential consequences for body size, as a shorter development time affords less time to accumulate body mass. Here, *20 lux* spiders were smaller at maturity, largely due to their earlier maturation. However, when maturation time was statistically controlled spiders (particularly females) reared under light at night were also smaller than *0 lux* spiders, suggesting that ALAN may directly affect growth rate in this species. This contrasts with a previous study in the cabbage moth which found no evidence that exposure to ALAN affected rates of body mass growth (*Van Geffen et al., 2014*), but such effects have received little attention across other invertebrate taxa. The length of the intermoult period in spiders is largely determined by rates of energy

accumulation and increases in body mass (*Nijhout, 2003*). ALAN may disrupt the body mass threshold for moulting and maturation to occur, resulting in lower increases in body mass and therefore a smaller body size at maturation. Further, artificial light potentially impaired their ability to consume food, or to convert food into body mass. Food consumption was kept as similar as possible between the two treatment groups, so a physiological disruption is the most likely explanation. The biological clocks that are regulated by melatonin in turn modulate feeding behaviours through effects on hormones that regulate hunger and satiety (*Challet, 2015*; *Fonken & Nelson, 2014*). These systems can also affect the physiological processing of food (*Challet, 2015*; *Fonken & Nelson, 2014*; *Fonken et al., 2010*). Hence, the impacts of ALAN on melatonin can cascade to fundamentally alter development and growth. To our knowledge, the effects of ALAN on these hormones have not been explicitly tested in arthropods, but analogues of some of these hormones have comparable physiological roles in insects (*Perić-Mataruga et al., 2009*).

A major cost of exposure to ALAN is an increase in mortality (*Eisenbeis & Hassel, 2000*; *Longcore & Rich, 2004*). Spiders exposed to ALAN exhibited higher lifetime mortality rates but, as ALAN also stimulated earlier maturation, the number of spiders that reached maturity and adult longevity were both comparable across the two groups. This increase in mortality may have been driven by increased oxidative stress, as is suggested for other species (*Jones et al., 2015*). Alternatively, it may have resulted from their accelerated development, as resources were diverted from growth and maintenance into reproductive development (*Boggs, 1992*). In contrast to previous studies (*McLay, Green & Jones, 2017*), there was no effect of ALAN on adult longevity. This can be explained by a trade-off between the deleterious effects of aging and the benefits of accruing more body reserves. Spiders in the *0 lux* treatment aged more prior to maturing, so their adult lifespan would be shortened; however, they also had greater body reserves, prolonging their adult lifespan. The high mortality we observed in both treatment groups may have been a result of the unnatural diet our spiders experienced, particularly as their diet lacked moths. However, this unnatural diet affected both treatment groups equally, so it is unlikely that it would dramatically alter our conclusions.

The impact of ALAN on the viability of urban spider populations depends on its effects on the reproductive fitness of individual spiders in a population. The *20 lux* spiders produced the same number of eggsacs but considerably fewer spiderlings compared to *0 lux* spiders. The reduction in fecundity for *20 lux* females was largely explained by ALAN induced differences in body mass, as similarly sized females produced comparable numbers of eggs regardless of light treatment. The lack of strong size-independent effects of ALAN on fecundity is contrary to previous results in *Drosophila* (*McLay, Green & Jones, 2017*). Nonetheless, our results suggest that a smaller body size due to ALAN exposure will lead to a reduction in reproductive fitness. There was no difference between treatment groups in adult longevity, so both groups had an equal amount of time for egg production. However, under natural conditions, predation and declining winter temperatures are major sources of mortality, so earlier maturation may extend the breeding period

(*Dominoni, Quetting & Partecke, 2013*; *Schneider & Elgar, 2002*). Hence, the ALAN-induced trade-off between maturing earlier or maturing larger also implies a trade-off between a longer breeding period or producing more spiderlings per eggsac for a shorter period.

In this study, we tested the effects of light treatment with both groups provided with the same amount of food, whereas spiders that build their webs near lights are likely to receive considerable foraging benefits (*Adams, 2000*; *Craig, 1987*). If greater food intake increases reproductive output (*Kreiter & Wise, 2001*; *Reed, Nicholas & Stratton, 2007*), the foraging benefits associated with artificial lights have the potential to at least partially compensate for the reproductive costs of this trade-off (*Lowe, Wilder & Hochuli, 2014*). This benefit may be tempered by observations of LED lights failing to attract moths—25 webs over three nights failed to catch moths, despite the observed presence of moths in the habitat (NJW, under review)—and decreasing moth capture rates in other orb-web spiders (*Yuen & Bonebrake, 2017*). Hence, the potential for foraging benefits to mask developmental costs depends on how ALAN affects the types of prey available to insectivores foraging around lights. We found no effect of light treatment on the mass of individual spiderlings. This has important potential fitness implications, as offspring size is related to offspring performance, including starvation tolerance and the ability to capture prey (*Walker, Rypstra & Marshall, 2003*). However, intergenerational effects may not be evident without further measurements of performance and development in subsequent generations.

## CONCLUSIONS

The combined direct and indirect effects of ALAN on urban insect populations depends on how shifts in life history patterns in these spiders affect predator–prey interactions for these affected spiders. Increased mortality and smaller body size may reduce the predation impact spiders exert on insect populations, counteracting increases in predation due to mutual attraction of both predator and prey towards artificial lights. ALAN is associated with high levels of urbanisation (*Elvidge et al., 2001*; *Hansen et al., 2001*; *Longcore & Rich, 2004*; *Ma et al., 2012*) and spiders living in urban habitats will experience not only the impacts of ALAN, but other environmental perturbations due to urbanisation. Understanding the impacts of urbanisation more generally for spider populations, and the likely consequences for insect communities, will require a more integrated consideration of these factors. Future research into the effects of ALAN on urban insectivores more generally should consider the impacts on both foraging success and development in individuals, and how these impacts translate into population-level effects.

## ACKNOWLEDGEMENTS

We thank Mark Elgar for his insightful advice and manuscript comments. We thank Lucy McLay, Caitlyn Perry and Po Peng for their help with spider maintenance, and Katrina-Lee Ware for counting the majority of the spiderlings.

### Funding

This research was supported by a grant from the Hermon Slade Foundation awarded to Therésa Jones (HSF 14/4) and an Australian Research Council grant to Therésa Jones (DP150101191). The funders had no role in study design, data collection and analysis, decision to publish, or preparation of the manuscript.

### Grant Disclosures

The following grant information was disclosed by the authors:
Hermon Slade Foundation awarded to Therésa Jones: HSF 14/4.
Australian Research Council grant to Therésa Jones: DP150101191.

### Competing Interests

The authors declare that they have no competing interests.

### Author Contributions

- Nikolas J. Willmott conceived and designed the experiments, performed the experiments, analyzed the data, contributed reagents/materials/analysis tools, prepared figures and/or tables, authored or reviewed drafts of the paper, approved the final draft.
- Jessica Henneken conceived and designed the experiments, performed the experiments, analyzed the data, authored or reviewed drafts of the paper, approved the final draft.
- Caitlin J. Selleck conceived and designed the experiments, performed the experiments, analyzed the data, authored or reviewed drafts of the paper, approved the final draft.
- Therésa M. Jones conceived and designed the experiments, analyzed the data, contributed reagents/materials/analysis tools, prepared figures and/or tables, authored or reviewed drafts of the paper, approved the final draft.

### Data Availability

  The raw data are provided in File S1.

### Supplemental Information

Supplemental information for this article can be found online at http://dx.doi.org/10.7717/peerj.5599#supplemental-information.

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
