# Peer review of "Artificial light at night alters life history in a nocturnal orb-web spider"

_PeerJ, doi:10.7717/peerj.5599_

## Round 0.1 · original submission · Minor Revisions

Dear Dr. Willmott and colleagues:

Thanks for submitting your manuscript to PeerJ. I have now received two independent reviews of your work, and as you will see, both are very favorable. Well done! Nonetheless, both reviewers raised some relatively minor concerns about the research, and areas where the manuscript can be improved. I agree with the reviewers, and thus feel that their concerns should be adequately addressed before moving forward.

Therefore, I am recommending that you revise your manuscript accordingly, taking into account all of the issues raised by the reviewers. I do believe that your manuscript will be ready for publication once these issues are addressed.

Good luck with your revision,

-joe

Reviewer 1 ·

Basic reporting

This is an interesting and thorough study, well described and performed. The manuscript is very well written throughout.
Literature coverage is appropriate.

Experimental design

Using a split-brood design, offspring from 18 females of an orb-web spider were subjected to two treatments that differed in the amount of light provided during 12hours of dark. Spiders either received 0 or 20 lux during the night and were raised from egg to adult. Growth and development of 440 individuals were carefully monitored and the number of moulting events was documented. Prey availability was standardised and increased twice throughout development to accommodate increasing demands.
The experiment was designed to test the hypothesis that artificial light at night (ALAN) would affect survival and life-history as well as female fecundity.

The experimental design varies a single variable (light at night) on two levels. A more complex design might have been valuable, as for example two levels of prey supply. Perhaps this could be planned as a follow-up?

Validity of the findings

Results show that the main effect of the treatment (light during the night) was an accelerated development with costs in adult body size and female fecundity. How well this result can be interpolated to a natural situation is unclear and I wonder whether the amount of prey provided was relatively lower than the average in nature. Perhaps that was intentional since an ad libitum provision of food might have masked the effects of ALAN? An explanation of the reasoning behind the applied feeding regime should be provided.

Additional comments

Despite a thorough description of experimental conditions, I have a few minor questions, none of which are serious concerns:
1. Eriophora is a nocturnal species. How did that affect web building and hunting behaviour? Did they perhaps take down webs during the day? Or did you observe any other differences during night and day, perhaps related to activity in some way? Any light-dependent activity could have affected prey capture directly, I think.
2. The light regime may have also influenced behaviour of the flying insect prey and possible differences in the flies’ activity may affect the probability of getting entangled into the web. Perhaps the authors have collected information about whether the provided prey items were actually eaten or not?
3. During each phase, a single prey species was provided, first Drosophila, then Musca and finally crickets. I am curious whether the provision was adequate to cover all needs for normal development. Could prey supply be part of an explanation for the high mortality rate? This is not a serious concern, as conditions were the same in both treatments but it might be relevant to briefly discuss whether the prey regime might have been rather poor or not despite the constant prey supply. There is no discussion about the appropriateness of prey quantity to simulate natural conditions and growth. Obviously, prey numbers were high enough to allow growth and moulting but did the duration of development and the size at maturation reflect natural conditions? Growth of almost a year seems a long time for a spider. Perhaps there are data on sizes and fecundity of wild females? If so, I would find a comparison interesting.
4. Adults received crickets matched to their size. Does that mean that smaller females received smaller crickets but the same number? If so, why was that done? Could they have eaten more if you would have given them more?

Reviewer 2 ·

Basic reporting

I think the article is very well written, clear and structured in an easy way that allows comparisons between the different sections of the manuscript, especially between methods, results and discussion. Two small points:

1. I think while the structure of methods, results and discussion is very similar and helped me assessing each component of your study (maturation time and survival, body mass, reproductive success), the same is not true for the statistical analyses section. I would sugest you to keep a clear distinction also in this section, clearly stating what were the models used for assessing changes in mat. time and survival, body mass and reproductive success. At present this section is a bit messy and it is not always clear which model was used for which analysis,

2. While I appreciate the long list of literature you cite and refer to, I think you have missed a key recent study in this same journal (Yuen and Bonebrake 2017 - Artificial night light alters nocturnal prey interception outcomes for morphologically variable spiders - Peer J). Here the authors showed something counterintuitive, spiderwebs in lit areas were less successfull in intercepting moths than webs in unlit areas, which is contrary to your expectations about potential benefits of artificial light at night for spiders' foraging success. I think that reporting the findings of this study in your own article would make it a much more unbiased and comprehensive discussion of your results.

Experimental design

The design is sound and well explained. Few minor questions/comments:

1. L126-141. Did all spiders mate? If not, was there any difference in the probability of mating between control and LaN spiders?

2. Related to the previous point. Did each individual mate only once or multiple times? If the latter, did reproductive success decrease with age, and was this different between light treatments? I am particularly interested in the possibility that spiders under LaN may have had an earlier onset of reproductive senescence.

3. L103-104. It's ok to report the lux range of the collection sites, but what about mean +- SD? I think this information is sorely needed. Based on these values, why did you choose 20 lux as your experimental light intensity? I think you need to specify this.

4. Relatd to the previous point. Any idea whether the light levels at the collection site influence any of your subsequent results? What proportion of the variance was explained by your random effect family in your mixed models?

5. L110. Were these plastic cups transparents? Otherwise how did you ensure that spiders were exposed to LaN?

6. L237-240. This aspect on growth trajectories is really interesting and made me thinking of whether you actually have the data to proper calculate growth curves of spiders in the different treatments, but reading your methods sections it's not clearly to me how many times body mass was measured per individual. Could you calculate growth curves and test for treatment effects in a proper way?

Validity of the findings

I think overall the data is robust and statistically sound, and the results well interpreted. I do have few comments though on the statistical approach and the way that results are reported.

1. L169-170. It's hard for the readers to appreciate the magnitude of the effect of LaN on longevity without clearly stating the effect size of this treatment compared to the control group. You state the effect size for the the reproductive success (L196), and that was really helpful to me to visualize the importance of this result. Pleae try to use the same approach throughout the results section and make use of effect sizes.

2. L177-178. This does not make much sense, as when a significant interaction is present one cannot really interpret the main effects. I suggest you delete the first part of this sentence and report only the result for the interaction term.

3. I question the approach of splitting the datasets into sexes and treatments. This way you are running a lot of independent models, increasing the risk of finding significant effects simply by chance. I would rather use posthoc tests with Bonferroni correction or any other way to correct for multiple testing, or bootstrapping methods that produce reliable confidence intervals upon which the comparison of different treatment groups and/or sexes can be done.

Additional comments

Although I have enjoyed reading your article and I think you present very interesting and strong results, I was also slightly disappointed in that I thought that you had the possibility of a much deeper insight into the effects of light at night on life-histories and fitness, by linking these data to more mechanistic/physiological aspects that might have helped you to explain your results. in the end, these data is very cool and convincing but I miss a clear explanation of how light at night affects maturation time, survival, body mass and reproductive success. Also, how likely is that spiders are exposed to such high levels of LaN in nature? 20 lux is dim, but still in the higher range of what animals can be exposed to in light polluted areas, particularly for mobile animals such as spiders. Also, the explanation for the earlier maturation time is not really comprehensive. What do the authors really think is happening? Do you think that spiders perceive a longer photoperiod, therefore being "tricked" into thinking that it is late in the season and they need to speed up their maturation if they want to be ready at the right time, or perhaps something more fundamental about the circannual and circadian rhythms of these animals is modified? For instance, see Wikelski et al 2008 Phil Trans B for a very interesting take into how different photoperiod might affect energy turnover which in turn might affect the speed of the circannual clock. in general, I think a more fundamental understanding of the mechanisms at play might have made this article much more compelling. But I do like it still!

---

## Round 0.2 · accepted · Accept

Dear Dr. Wilmott and colleagues:

Thanks for further revising your manuscript based on the minor concerns raised by the reviewers. I now believe that your manuscript is suitable for publication. Congratulations! I look forward to seeing this work in print, and I anticipate it being an important resource for spider biology and ecology. Thanks again for choosing PeerJ to publish such important work.

Best,

-joe

#